# Repurposing Antidepressants and Phenothiazine Antipsychotics as Efflux Pump Inhibitors in Cancer and Infectious Diseases

**DOI:** 10.3390/antibiotics12010137

**Published:** 2023-01-10

**Authors:** Bálint Rácz, Gabriella Spengler

**Affiliations:** Department of Medical Microbiology, Albert Szent-Györgyi Health Center, Albert Szent-Györgyi Medical School, University of Szeged, Semmelweis utca 6, 6725 Szeged, Hungary

**Keywords:** drug repurposing, MDR efflux pumps, multidrug resistance, antidepressant, antipsychotic, selective serotonin reuptake inhibitors (SSRIs), tricyclic antidepressants (TCAs), phenothiazines

## Abstract

Multidrug resistance (MDR) is a major obstacle in the therapy of infectious diseases and cancer. One of the major mechanisms of MDR is the overexpression of efflux pumps (EPs) that are responsible for extruding antimicrobial and anticancer agents. EPs have additional roles of detoxification that may aid the development of bacterial infection and the progression of cancer. Therefore, targeting EPs may be an attractive strategy to treat bacterial infections and cancer. The development and discovery of a new drug require a long timeline and may come with high development costs. A potential alternative to reduce the time and costs of drug development is to repurpose already existing drugs. Antidepressants and antipsychotic agents are widely used in clinical practice in the treatment of psychiatric disorders and some somatic diseases. Antidepressants and antipsychotics have demonstrated various beneficial activities that may be utilized in the treatment of infections and cancer. This review aims to provide a brief overview of antibacterial and anticancer effects of selective serotonin reuptake inhibitors (SSRIs), tricyclic antidepressants (TCAs) and phenothiazine antipsychotics, while focusing on EPs. However, it should be noted that the antimicrobial activity of a traditionally non-antibiotic drug may have clinical implications regarding dysbiosis and bacterial MDR.

## 1. Introduction

Drug repurposing (or drug repositioning, reprofiling, or re-tasking) is a strategy to find new applications for approved drugs that are outside the scope of the original medical indication [1,2]. The strategy of drug repositioning or repurposing rules out the structural modification of the drugs, it means a new indication regarding the biological properties or therapeutic use such as a new formulation, a new dose or via a new route of administration [2]. There are several advantages of drug repurposing, such as a lower risk of failure and better safety profile, shorter drug development procedures because of the already completed preclinical tests, and lower financial costs [2]. The repositioning of drugs could be an attractive alternative to treat human diseases such as cancer, infectious diseases, neurodegenerative and autoimmune diseases [3]. The approach has to face several challenges, including the lack of knowledge on regulatory requirements, financial and resource difficulties, new problems in clinical trials resulting in missing or inadequate proof, intellectual property issues that can hamper the commercialization of the given drug, and market analysis [4].

Microorganisms naturally develop antimicrobial resistance as a result of interactions with their environment [5]. The number of infections caused by multidrug resistant (MDR) bacteria is increasing as a consequence of selection pressure from the widespread use of antibiotics [6]. However, the MDR phenotype is not unique to microbes, since MDR cancer is a serious hindrance to oncological treatment leading to relapses and recurrences [7]. There are similarities in the drug resistance mechanisms of bacteria and cancer cells. Drug resistance in bacteria and cancer may be intrinsic or acquired, and may involve the inactivation of the drug, overexpression of efflux pumps (EPs), and modification of drug targets [8,9]. The role of EPs in bacteria and cancer is more than just detoxification. In bacteria, EPs may mediate cell–cell communication and biofilm formation, thus enabling bacteria to adapt to various environmental stimuli and contributing to the virulence of bacteria [10]. In cancer, EPs may promote cancer growth not only by extruding anticancer drugs but also by other mechanisms that are related to cell proliferation and metastasis [11].

Antidepressants, especially selective serotonin reuptake inhibitors (SSRIs), are widely prescribed in clinical practice to treat mood and anxiety disorders. In cancer patients, besides treating symptoms of depression and anxiety, they can be used as adjuvants in the therapy of neuropathic pain; additionally, SSRIs can alleviate hot flashes caused by hormone therapy [12]. Antipsychotics in oncology are also essential in supportive therapy for managing cancer-related delirium and other psychotic disorders [12]. Moreover, they may be preferred to benzodiazepines to treat acute anxiety and insomnia, since benzodiazepines may be delirogenic and cause respiratory depression [12]. The antiemetic effects of antipsychotics can be beneficial in the treatment of chemotherapy-induced nausea and the side effects of increased appetite and weight gain can help in cancer-related anorexia [12]. Numerous studies have emerged indicating that psychiatric drugs may have anti-inflammatory, antimicrobial, anticancer properties, and they may be attractive candidates to combat bacterial multidrug resistance and cancer (Figure 1) [13,14,15]. In this review, we aim to present an overview of the antibacterial, anticancer, and efflux pump modulating activity of SSRIs, tricyclic antidepressants (TCAs), and phenothiazine-type antipsychotics (Figure 2 and Figure 3).

## 2. Antibacterial Resistance

Drug resistance is a natural process and is based on the interaction of organisms with their environment [16]. However, during the last decades, the occurrence of drug-resistant microbial infections has increased dramatically which can be explained by the global spread of drug resistant microbes (bacteria, fungi, viruses, and parasites) and the extensive use and misuse of antimicrobial agents. The spread of antibiotic resistance is related to high mortality rates and high costs regarding the therapy and hospitalization of patients [17].

Concerning the antibiotic resistance of bacteria, there are natural (intrinsic) and acquired (extrinsic) resistance mechanisms. In the former case, the cells can be considered to be resistant even before encountering the harmful agents and this resistance is a trait of a species or genus. In contrast, the acquired resistance arises from mutations and horizontal gene transfer [16].

There are numerous antibiotic resistance mechanisms such as restricted penetration, drug efflux, target modification, destruction/modification of the antibiotics, target switching, and target sequestration [18]. Out of these mechanisms, the overexpression of the efflux pump can result in higher virulence of bacteria because these pumps can contribute to bacterial communication (quorum sensing) and biofilm formation [10].

### 2.1. Bacterial Efflux Pumps

Bacterial efflux pumps have the capacity to regulate the internal environment by removing harmful agents, metabolites as well as cell–cell communication signal molecules. The expression of efflux pumps is precisely regulated by different local and global transcriptional regulators suggesting that these pumps have essential physiological functions. Concerning the physiological functions, efflux pumps are crucial in stress adaptation, pathogenesis, and virulence of bacteria. The over-expression of multidrug efflux pumps has been increasingly found to be associated with clinically relevant drug resistance [19]. There are six EP families based on their membrane topology, energy coupling mechanisms as well as substrate specificities. These EP families are as follows: ATP-binding cassette (ABC) family, multidrug and toxic compound extrusion (MATE) family, small multidrug resistance (SMR) family, major facilitator superfamily (MFS), resistance nodulation division (RND) family, and proteobacterial antimicrobial compound efflux (PACE) family [19].

### 2.2. Bacterial Efflux Pump Inhibitors (EPIs)

The abolishment of efflux mechanisms could be achieved in different ways: (1) the downregulation of the expression of EP genes, (2) the re-design of antibiotics and development of new antibiotics, (3) the inhibition of the assembly of functional pumps, (4) the inhibition of the substrate binding by competitive or non-competitive inhibitors, (5) blocking the outer membrane channel, as well as (6) interference with the energy supply of pumps [20].

Efflux pump inhibitors alone could sensitize bacteria to antibiotics before antibiotic treatment or in combination they could increase the susceptibility of resistant strains by increasing the intracellular concentration of antibiotics. EPIs alone or in combination with conventional antibiotics are able to block multidrug efflux systems. The first EPI compound MC-207,110 (Phe-Arg-β-napthylamide or PAβN) was described in 2001 that could inhibit the clinically relevant efflux pumps of *Pseudomonas aeruginosa*; furthermore, it could inhibit other RND-type pumps of Gram-negative bacteria [21].

Classification of EPIs is a difficult task because some inhibitors are pump specific, while others are not and have multiple targets in the bacterial cells. Based on their origin, bacterial EPIs can be classified into two major groups: natural bioactive agents and synthetic compounds [22]. However, drug development is a time-consuming and expensive way to find new EPIs. An auspicious strategy could be the drug repurposing approach that applies existing drugs in a new indication, in this case, to treat infections. This perspective has numerous advantages such as lower costs and a short development period [21].

#### 2.2.1. Selective Serotonin Reuptake Inhibitors (SSRIs) and Serotonin and Norepinephrine Reuptake Inhibitors (SNRIs)

It has been demonstrated that selective serotonin reuptake inhibitors (SSRIs) possess antimicrobial activity [23]. According to Kaatz et al., phenylpiperidine-SSRIs exerted mild intrinsic antimicrobial activity against *Staphylococcus aureus*, *P. aeruginosa*, and *Escherichia coli* expressing MDR efflux pumps. One isomer of *paroxetine* showed EPI activity against MFS-(NorA and TetK) and RND-class (AcrB) efflux pumps (Figure 4) [24,25].

Sertraline is able to inhibit bacterial growth and the activity of efflux pumps in *E. coli*; furthermore, increased expression of *marA* and *acrB* was detected after sertraline treatment. It can be concluded that sertraline could influence the activity of efflux pumps and the expression of MDR and efflux-related genes; however, the plasma concentration in the patients is below the concentrations that is required for efflux pump inhibition (10–150 ng/mL in patients versus 30.6 μg/mL in the assay) [26].

Fluoxetine treatment was effective against the biofilm formation of the urinary tract pathogen *Proteus mirabilis*. It is known that efflux pumps are essential for biofilm production, and they can promote the colonization of *P. mirabilis*. Molecular docking results confirmed that fluoxetine attaches to the channel region of the biofilm-associated Bcr/CflA transporter (Figure 4) [27].

Citalopram acted synergistically with antibiotics against Gram-positive and Gram-negative bacterial strains. Resistance of *E. coli* to cefixime and *P. aeruginosa* to cloxacillin was reversed in the presence of citalopram by disk diffusion method. A total of 310 µg/mL of citalopram combined with levofloxacin, moxifloxacin, and gentamicin enhanced the activity of these antibiotics against *S. aureus* [28].

Fluvoxamine exerted potent antibacterial activity against *E. coli*, *Klebsiella pneumoniae*, *Acinetobacter baumanii*, and *Staphylococcus epidermidis* by disc diffusion assay as described in the study of Kalaycı et al., who tested the antimicrobial activity of various psychotropic drugs against bacteria [29]. However, it should be pointed out that the chronic use of antidepressant medication can influence the gut microbiota leading to potential adverse effects [30].

The widely used antidepressant duloxetine acted synergistically with chloramphenicol in *E. coli* contributing to the evolution and spread resistant *E. coli* clones. The expression levels of *acrA, acrB*, and *marA* genes were upregulated 1.7–2.2-fold by combined duloxetine and chloramphenicol treatment compared to the exposure to the single drugs. With the enhancement of the AcrAB-TolC system, multidrug resistant *E. coli* clones can evolve and cause the emergence of resistant strains in the gut. Since the gut microbiota may have been exposed to non-antibiotic pharmaceuticals and antibiotics for a longer period of time, bacteria can develop resistance towards these agents in vivo [31].

Using the disk diffusion technique, 600 µg/mL of venlafaxine improved the activity of ciprofloxacin, levofloxacin, norfloxacin, and moxifloxacin against *P. aeruginosa*. It is suspected that venlafaxine can inhibit bacterial efflux pumps and can compete with the substrates of these pumps [28].

#### 2.2.2. Tricyclic Antidepressants (TCA)

The interaction of amitriptyline and norfloxacin was synergistic against *Salmonella* Typhimurium [32]. Furthermore, it has been confirmed that amitriptyline can potentiate the activity of AcrB substrates. In combination with antibiotics, amitriptyline increased the activity of chloramphenicol, nalidixic acid, and tetracycline against *S.* Typhimurium overexpressing the AcrAB-TolC efflux system. In addition, besides its intrinsic antimicrobial activity, amitriptyline also augmented the susceptibility of *A. baumannii* to chloramphenicol. It was confirmed that amitriptyline is an AcrB substrate, and it is able to bind to residues of AcrB that are crucial for substrate recognition and/or transport via AcrAB-TolC system [33]. Imipramine had a slight antibacterial effect against methicillin-susceptible and -resistant *S. aureus*; however, the concentrations applied in the assay cannot be used in patients (the therapeutic concentration is 175–300 ng/mL versus the bacterial MIC of 128 μg/mL) [34]. The tricyclic psychopharmacons imipramine and desipramine possessing non-planar tricyclic skeletons become relatively inactive on saturation of the ring system regarding their antibacterial effect. Furthermore, imipramine and desipramine were able to eliminate the F’lac plasmid from *E. coli* K12 strain and in this aspect, these compounds could reverse the spread of resistance mediated by plasmids [35].

#### 2.2.3. Phenothiazines

Many of the phenothiazines possess wide-ranging antibacterial activity against *Mycobacteria*, some Gram-positive and Gram-negative bacteria [36]. Regarding the mode of action, phenothiazines can target the bacterial membrane, the nucleic acids, and the energy supply of the pumps. Since these compounds may also be AcrB substrates, their interaction is not selective [36]. The application of a subinhibitory concentration of a non-antibiotic substance such as phenothiazines can enhance the susceptibility of bacteria towards antibiotics indicating that efflux mechanisms are involved in this phenomenon (Figure 4) [37].

Thioridazine (TZ), chlorpromazine (CPZ), and fluphenazine reduced NorA-mediated ethidium bromide efflux by at least half indicating that these phenothiazines are able to inhibit NorA of *S. aureus* and other non-NorA-mediated resistance mechanisms in a dose-dependent manner [38]. In addition, it was confirmed that in the case of MRSA, the susceptibility to oxacillin was influenced by CPZ or TZ probably due to efflux-related mechanisms [39]. In the case of *Burkholderia pseudomallei,* the intrinsic resistance to aminoglycosides and macrolides is due to the RND efflux pumps BpeAB-OprB and AmrAB-OprA [40], which are secondary pumps deriving their energy from the proton motive force (PMF). The combined administration of chlorpromazine or prochlorperazine with erythromycin could inhibit the efflux of erythromycin by disrupting the proton gradient required for the function of the efflux pumps. These results may confirm the potential use of phenothiazines as EPIs and resistance modifiers in *B. pseudomallei*; however, the high concentrations (250 μM to 1 mM or 79.715 μg/mL to 318.86 μg/mL) applied in the experiments cannot be achieved in patients (therapeutic concentration: 30–300 ng/mL) [41]. Chlorpromazine treatment provoked an increase in the expression of *ramA*, and a reduction in the expression of *acrB* in *Salmonella* Typhimurium [32].

TZ and CPZ increased the intracellular accumulation of the efflux pump substrate ethidium bromide in *Mycobacterium smegmatis* and *Mycobacterium avium*. It has been confirmed that thioridazine could inhibit the intrinsic efflux pump system of *M. avium* causing erythromycin resistance [42]. CPZ and TZ exerted a direct inhibitory effect on respiration in *Mycobacterium tuberculosis* as shown by transcriptional responses detected by microarray [43]. Following TZ treatment, the gene of the efflux pump Mmr was upregulated in *M. tuberculosis* [44]. It was confirmed that CPZ being an AcrB substrate could enhance the intracellular accumulation of ethidium bromide and norfloxacin in *S*. Typhimurium. According to molecular docking studies, CPZ has a binding site beneath the CH3 channel of the AcrB pump and the binding of CPZ can interfere with some conformational states of AcrB during substrate efflux indicating that CPZ could be an inhibitor of the pump [33].

CPZ could inhibit the chloramphenicol resistance in *Enterobacter aerogenes* isolates reversing the resistance of these strains that was indicated for example by a 32-fold reduction in the original MIC of chloramphenicol in the presence of CPZ [45].

The EPI properties of promethazine (*PMZ*) are pH-dependent because the EPI activity of PMZ was less effective at acidic pH compared to a neutral pH in *E. coli* K12 AG100 because the proton motive force can drive the AcrAB-TolC system more efficiently at acidic pH. After 18 h of PMZ treatment, the efflux pump genes *acrA* and *acrB* were upregulated at pH 5 and pH 7 confirming the defense mechanism of bacteria in order to remove the noxious agents from the cells (Figure 4) [46]. In *Burkholderia pseudomallei,* PMZ increased the susceptibility to erythromycin, trimethoprim/sulfamethoxazole, gentamicin, and ciprofloxacin. In addition, PMZ could alter the structure of bacterial biofilm improving the penetration of antibiotics by damaging the biofilm structure (Figure 4) [47].

### 2.3. Potential Risks: Dysbiosis and Antibiotic Resistance

SSRIs and several other psychotropic drugs possess antibiotic effects, that can have direct consequences for the composition and stability of the gut microbiome [23,48]. Some SSRIs may exert antimicrobial activity against gut microbes for several hours each day, such as sertraline, fluoxetine, paroxetine, and fluvoxamine, and the different species and strains differ in their susceptibility to SSRIs. It was confirmed that sertraline and fluoxetine may exert stronger inhibitory effects on Gram-positive bacteria than on Gram-negative bacteria [49]. It is obvious that SSRIs influence the balance of the gut microbiome and microbial defense system including bacterial efflux pump systems, motility of microbes; furthermore, these drugs may have a synergistic interaction with other drugs provoking dysbiosis [48].

Psychoactive drugs can trigger the SOS response in Gram-negative bacteria influencing the virulence of these strains [50]. It should be pointed out that SSRIs are detectable in wastewater samples, and in surface waters [51,52]; furthermore, the accumulation of SSRIs in animals can appear in the food chain as demonstrated in fishes [52].

## 3. Anticancer Activity of SSRIs, TCAs, and Phenothiazines

SSRIs, TCAs, and phenothiazines achieve their anticancer activity through various mechanisms, including the modulation of different signaling pathways (e.g., NFκB, AKT, mTOR, β-catenin), altering membrane permeability and cellular metabolism, inducing endoplasmic reticulum (ER) stress and modulating autophagy, influencing intracellular Ca^2+^, ultimately resulting in sensitization to apoptotic cell death and antimetastatic effects (Table 1, Table 2 and Table 3, Figure 5).

### 3.1. Modulation of Signal Pathways

NFκB signaling has a complex role in inflammation and cellular proliferation, since it may have proinflammatory, tumorigenic and anti-inflammatory, anti-tumorigenic roles. However, the modulation of the NFκB pathway may be an effective therapeutic target in some cancers [95]. AKT is a proto-oncogene overexpressed in different cancers, it has a regulatory function in cell proliferation and apoptosis. The downstream effector of AKT mTOR has many different roles in cellular functions, it is involved in glycolytic and lipid metabolism, suppression of autophagy, growth factor receptor signaling and neoangiogenesis, and promoting tumor growth and metastasis formation. Inhibiting the aforementioned pathways is a promising approach in cancer therapy [96,97]. Numerous studies have shown the modulation of these pathways in different cancerous cell lines when treated with SSRIs, TCAs, and phenothiazines, leading to apoptosis induction, autophagy modulation, and antimigratory effects (Table 1, Table 2 and Table 3). Sertraline and thioridazine were also able to inhibit translationally controlled tumor protein (TCTP), a regulator of cell proliferation associated with the mTOR pathway [98,99,100]. Additionally, sertraline, amitryptiline, and TZ inhibited β-catenin signaling, which has many protumorigenic roles and is linked to cancer stem cells [101,102,103].

### 3.2. Effects on Cellular Metabolism

The effect of antidepressants and antipsychotics on mitochondria and cellular metabolism is one of the key mechanisms by which they exert their anticancer effects. The administration of SSRIs, TCAs, phenothiazines promotes ROS generation and leads to mitochondrial dysfunction by disrupting mitochondrial membrane potential, which results in disrupted cellular respiration and cytochrome c leaking, ultimately causing apoptosis (Table 1, Table 2 and Table 3). Furthermore, fluoxetine has also inhibited sphingomyelin phosphodiesterase 1, causing sphingomyelin accumulation and cell death in glioblastoma cell lines [104]. In BRAF/MEK inhibitor-tolerant melanoma persister cells fatty acid oxidation is essential to fuel cellular metabolism, this may be exploited by TZ, since it can suppress peroxisomal β-oxidation [105]. Interestingly, paroxetine could reduce doxorubicin-mediated cardiotoxicity in male Wistar rats that is related to free radical formation as a result of doxorubicin metabolism [106].

### 3.3. Ca^2+^ Overload

Ca^2+^ is a key second messenger responsible for essential cellular processes in both normal cells and cancerous cells, such as metabolism, proliferation, and cell death. Uniquely intracellular Ca^2+^ level, unlike other second messengers, is regulated by compartmentalization orchestrated by Ca^2+^ channels and transporters of the cell membrane and Ca^2+^ storing organelles, namely the ER and mitochondrium [107]. SSRIs, TCAs, and phenothiazines may increase cytosolic free Ca^2+^ by liberating Ca^2+^ from intracellular stores or causing Ca^2+^ influx from extracellular space that may induce mitochondrial dysfunction, ER stress, and apoptosis in various cancerous cell lines (Table 1, Table 2 and Table 3).

### 3.4. Drug Efflux Pumps in Cancer

One of the main functions of ABC-transporters is to protect the organism against potentially harmful molecules and thus ABC-transporters are mainly expressed at barrier surfaces, e.g., blood–brain barrier, gastrointestinal and hepatobiliary tract, renal tubules [108]. In cancer, ABC-transporters, besides extruding chemotherapeutic drugs, may be involved in apoptosis regulation, cell migration [109,110]. Some substrates of ABC-transporters, such as cyclic nucleotides, platelet-activating factors, prostaglandins, leukotrienes, and cholesterols, are also significant regarding tumor metabolism and progression [11]. Seven subfamilies belong to the ABC superfamily; among them, three members of subfamilies are commonly involved in MDR cancer: P-glycoprotein (P-gp), multidrug resistance protein 1 (MRP 1 or ABCC1), and breast cancer resistance protein (BRCP or ABCG2) [111].

P-gp or ABCB1 was thought to be the most important mechanism responsible for MDR. According to the cancer stem cell (CSC) concept, there are CSC subpopulations in the cancerous tissue that are inherently resistant to chemotherapeutics by their ability to express efflux pumps, repair their DNA, and remain quiescent [112,113]. P-gp is coded by the *MDR1* gene; furthermore, the transcriptional regulation of *MDR1* gene expression is mediated by various signal pathways. The activation of NFκB, AKT, and Wnt/β-catenin signaling may enhance the transcription of *MDR1* gene which may lead to the increased expression of P-gp. Therefore, targeting these pathways may be an effective approach to increase the sensitivity of MDR cancer [114]. Another effective approach may be targeting tumor metabolism, since cancerous cells adapt their metabolism to fuel their metabolic needs for rapid proliferation. In cancerous cells, a high rate of glycolysis can be observed to provide precursors for the anabolic pathways, still ATP, which is needed for proliferation and ATP-consuming efflux pumps, is mainly generated by mitochondria. In addition, mitochondrial respiration is essential for generating reactive oxygen species (ROS) for cell proliferation [115]. To counterbalance the increased ROS, cancer cells have higher levels of ROS-scavengers; however, this balance is fragile and modulating ROS may be an effective approach to combat cancer and drug resistance [116,117].

### 3.5. Targeting Efflux Pumps in Cancer with SSRIs, TCAs, and Phenothiazines

Antidepressant and antipsychotic drugs may be promising candidates against P-gp-expressing cancers, since they may reduce the transcription of the *MDR1* gene by inhibiting the signal pathways that regulate its transcription and the inhibition of mitochondrial respiration may deprive tumor cells of ATP, which is essential for ABC-transporters. Several in vitro and in vivo studies also suggest that antidepressants and antipsychotics are direct inhibitors or substrates for P-gp (Figure 1, Figure 2 and Figure 3) [118,119,120].

Fluoxetine, a widely used SSRI antidepressant with a broad safety range, is emerging as a new chemosensitizer in preclinical models. In vivo studies showed prolonged survival of nude mice bearing P-gp expressing resistant MCF-7/ADM breast cancer and HCT-15 colon cancer, when treated with fluoxetine and doxorubicin [121,122]. Fluoxetine treatment resulted in the downregulation of P-gp protein expression. When combined with paclitaxel, it downregulated the *MDR1* gene expression in MCF7/ADM human breast cancer cell line; moreover, cytosolic glutathione S-transferase was also downregulated, which is a commonly expressed antioxidant system in resistant cancer [123,124]. Sertraline, another SSRI with halogenic atoms, demonstrated promising efflux pump inhibitory activity in P-gp expressing OVCAR-8 human ovarian cancer in vitro and in vivo xenograft model, combined with doxorubicin or pegylated liposomal doxorubicin [125].

Thioridazine, a first-generation phenothiazine antipsychotic that was widely used in psychotic disorders, could sensitize antimitotic drug resistant KBV20C oral cancer cells to vinblastine, although its potential use in cancer therapy might be limited by its cardiac toxicity [126,127]. On the same cell line fluphenazine, another phenothiazine antipsychotic included on the WHO Model List of Essential Medicines, achieved similar chemosensitizing effects [128,129]. Fluphenazine also decreased the expression of *MDR1 (ABCB1)* and *COX2* genes in doxorubicin-resistant LoVo/Dx colon adenocarcinoma cell lines. Moreover, the combination of fluphenazine and simvastatin could enhance the COX-2 inhibitory activity and sensitivity to doxorubicin [130]. Promethazine, a phenothiazine derivative, mainly used as an antihistamine and antiemetic agent, exhibited chemosensitizing activity in MCF7 human breast cancer drug resistant sublines when combined with vincristine or doxorubicin by modulating P-gp activity and decreasing the expression of *MDR1* and *MRP1* genes [131,132].

## 4. Conclusions

The rising microbial drug resistance is a public health threat worldwide and multidrug resistance in cancer is a major barrier to chemotherapeutic treatment. Drug repositioning may be an attractive strategy to combat multidrug resistance in bacteria and cancer, since it may be a shorter procedure and less costly than traditional drug development. Efflux pump-mediated resistance is one of the major mechanisms contributing to MDR, therefore targeting EPs may be an attractive approach that may improve the efficacy of chemotherapeutics in MDR bacteria and cancer. Furthermore, inhibiting efflux pumps may decrease bacterial virulence, and in cancer, it may inhibit metastasis formation and decrease relapses, since EPs are also involved in other processes than detoxification.

Numerous studies have emerged describing the antibacterial and anticancer effects of antidepressant and antipsychotic drugs. Many of the SSRIs and some TCAs showed promising activity regarding efflux pump inhibition in various Gram-positive and Gram-negative bacteria. Phenothiazine antipsychotics have also shown anti-mycobacterial activity and EP inhibition in mycobacteria. Regarding anticancer activity of SSRI, TCA, and phenothiazine antipsychotics, several in vitro and in vivo studies showed anticancer activity, which is associated with signal pathway modulation, cellular metabolism alteration, Ca^2+^-overload. Additionally, this may also lead to EP modulation, although several antidepressants and antipsychotics are direct EP inhibitors or substrates of EPs. Exploring EP inhibitor activity of antidepressants and antipsychotics in MDR cancers may be important since they are already used for supportive therapy in cancer patients.

While repurposing antidepressants and antipsychotics as antibacterial or anticancer agents may be an attractive idea, there are some risks that should be taken into consideration. In some cases, to achieve the antibacterial, anticancer, or EP inhibitory effects higher doses should be given. SSRIs have a wide therapeutic window, but the application of TCAs and phenothiazine antipsychotics may be limited due to their cardiac toxicity. Another question that should be taken into account regarding the antibacterial activity of antidepressants and antipsychotics is how they can contribute to the rapid spread of drug-resistant bacteria. Mental health disorders, e.g., depression, are of great importance. This may be explained by increasing economic and social inequalities and public health emergencies, thus the demand for antidepressants is increasing, which may also increase the selection pressure on bacteria [133,134]. Another problem is that some patients with depression and schizophrenic patients require lifelong treatment with psychiatric drugs. This medication may alter the gut microbiota also contributing to the development of metabolic side effects of these drugs, resulting in increased risk for metabolic diseases [135]. This may suggest that there might be a need to use pro- or prebiotics, when taking traditionally non-antibiotic drugs with antibacterial effects [136].

## Figures and Tables

**Figure 1 antibiotics-12-00137-f001:**
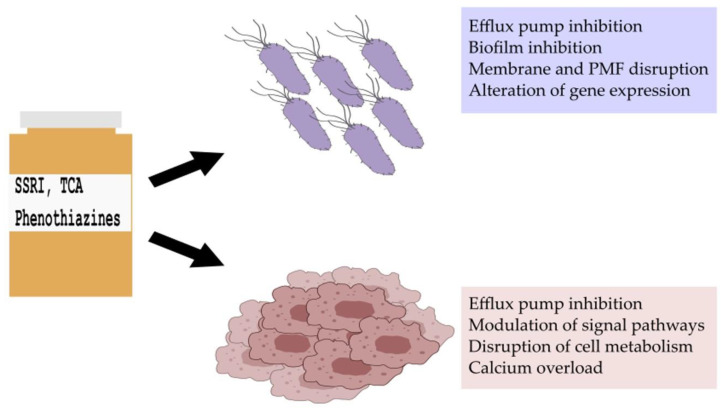
Antibacterial, anticancer, and efflux pump-modulating activity of selective serotonin reuptake inhibitors (SSRIs), tricyclic antidepressants (TCAs), and phenothiazine-type antipsychotics. PMF: proton motive force.

**Figure 2 antibiotics-12-00137-f002:**
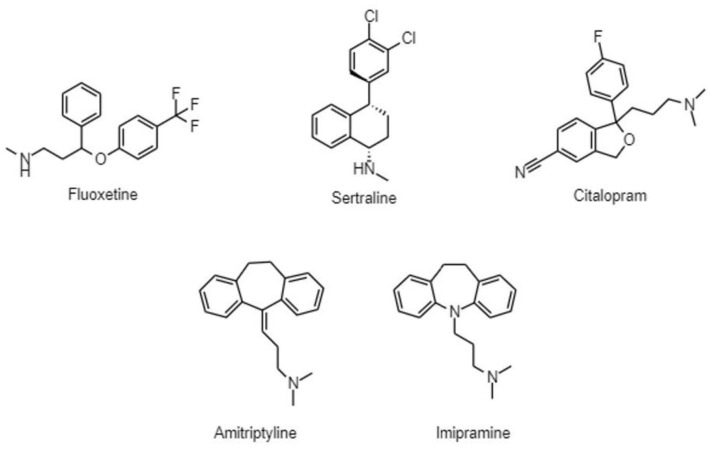
Chemical structures of antidepressants.

**Figure 3 antibiotics-12-00137-f003:**
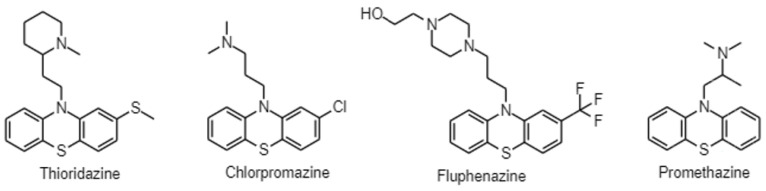
Chemical structure of phenothiazines.

**Figure 4 antibiotics-12-00137-f004:**
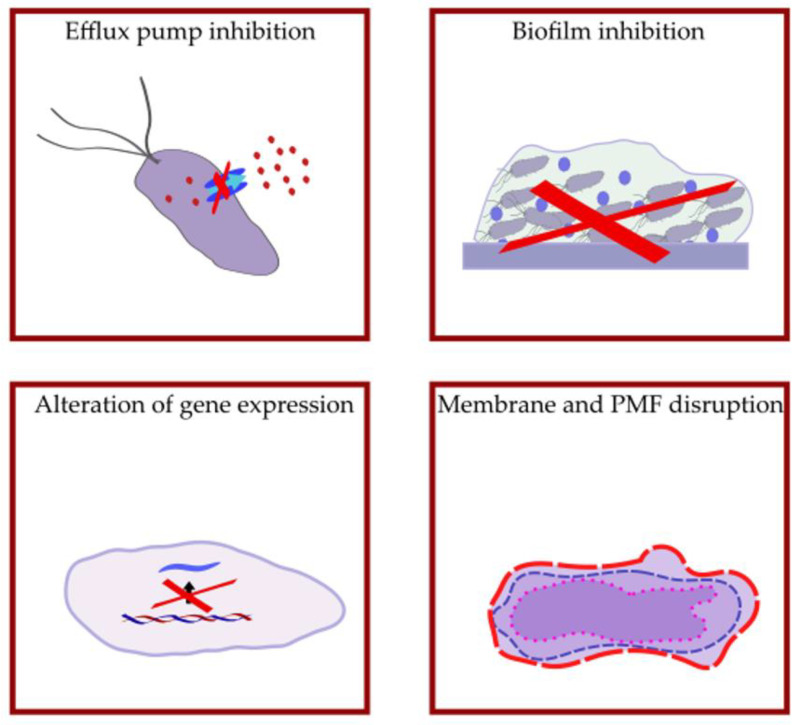
Antibacterial activity of selective serotonin reuptake inhibitors (SSRIs), tricyclic antidepressants (TCAs), and phenothiazine-type antipsychotics. PMF: proton motive force.

**Figure 5 antibiotics-12-00137-f005:**
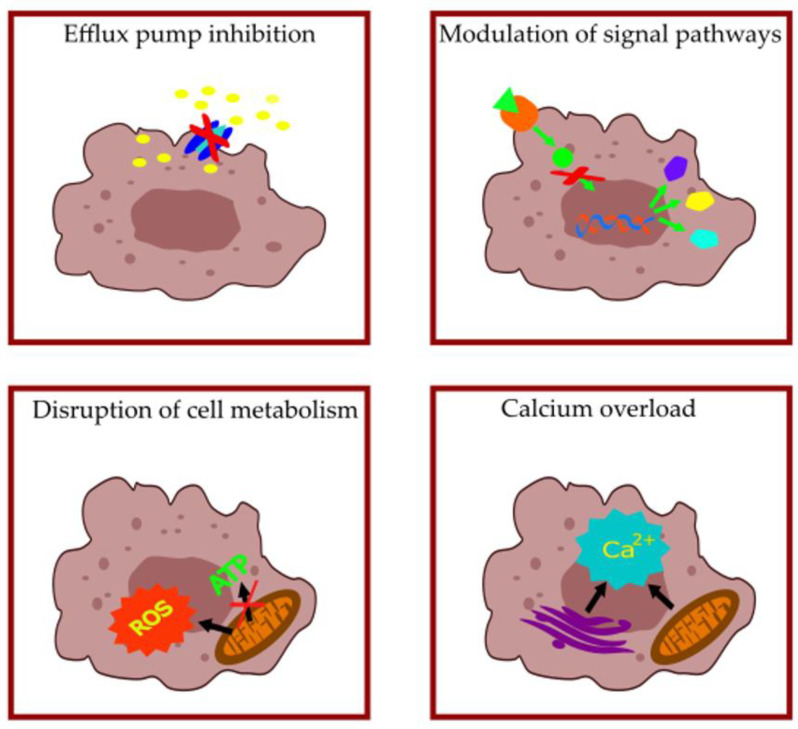
Main mechanisms of anticancer activity of selective serotonin reuptake inhibitors (SSRIs), tricyclic antidepressants (TCAs), and phenothiazine-type antipsychotics.

**Table 1 antibiotics-12-00137-t001:** Selective serotonin reuptake inhibitors (SSRIs) with anticancer properties.

Compound	Mechanism of Action	Cell Line	Reference
Citalopram Escitalopram	Modulation of NFκB-signaling, ROS formation	HepG2 hepatocellular carcinoma cell line, A549 and H460 non-small lung cancer cell lines	[53,54]
	Disruption of mitochondrial membrane potential	Burkitt lymphoma cell lines	[55]
Fluoxetine	Modulation of NFκB-signaling	CL1-5-F4 human lung adenocarcinoma cell line	[56]
		OVCAR-3 human epithelial ovarian cancer cell line	[57]
	Modulation of AKT/mTOR pathway	A549 and H460 non-small lung cancer cell lines	[58]
	Disruption of mitochondrial membrane potential	HT29 and CaCo-2 human colon adenocarcinoma cell lines, Burkitt lymphoma cell lines	[56,59]
	Mitochondrial Ca^2+^-overload	HeLa human cervical carcinoma cell line	[60]
Paroxetine	Modulation of AKT-signaling	HCT116 and HT29 human colon adenocarcinoma	[61]
	Disruption of mitochondrial membrane potential	Burkitt lymphoma cell lines	[55]
Sertraline	Modulation of mTOR signaling	MCF-7 human breast adenocarcinoma cell line	[62]
		SGC-7901/DDP gastric cancer cell line	[63]
	Modulation of AKT signaling	A375 human melanoma cell line	[64]
	Ca^2+^-overload, ROS formation	Prostate cancer cell lines	[65,66]
		MG63 human osteosarcoma cell line	[67]

**Table 2 antibiotics-12-00137-t002:** Tricyclic antidepressants (TCAs) with anticancer properties.

Compound	Mechanism of Action	Cell Line	Reference
Amitryptiline	Modulation of NFκB-signaling	T98G human glioblastoma multiforme cell line	[68]
	Inhibition of mitochondrial respiration	IPSB-18 anaplastic astrocytoma-derived cell line	[69]
		SK-MEL28, SK-ML2 and patient-derived melanoma cell lines	[70]
Nortryptiline	Inhibition of mitochondrial respiration	SK-MEL28, SK-ML2 and patient-derived melanoma cell lines	[70]
Imipramine	Modulation of NFκB-signaling	T98G human glioblastoma multiforme cell line	[68]
		CL1-5-F4 human lung adenocarcinoma cell line	[71]
	Modulation of AKT/mTOR pathway	U-87MG glioma cells	[72]
	Modulation of AKT- and NFκB-signaling	PC-3 human prostate cancer cell line	[73]
	Inhibition of mitochondrial respiration	IPSB-18 anaplastic astrocytoma-derived cell line	[69]
Clomipramine	Inhibition of mitochondrial respiration	IPSB-18 anaplastic astrocytoma-derived cell line	[69]
		SK-MEL28, SK-ML2 and patient-derived melanoma cell lines	[70]
Desipramine	Ca^2+^-overload	Hep3B hepatocellular carcinoma	[74]
		MG63 human osteosarcoma cell line	[75]

**Table 3 antibiotics-12-00137-t003:** Phenothiazines with anticancer properties.

Compound	Mechanism of Action	Cell Line	Reference
Thioridazine	Modulation of mTOR-signaling	ECA-109 and TE-1 esophageal squamous cell carcinoma	[76]
		Human cervical and endometrial cancer cell lines	[77]
	Modulation of AKT-signaling	A549 stem cell-like non-small lung cancer cell lines	[78]
		HepG2 hepatocellular carcinoma cell line	[79]
		Caki human renal carcinoma cell line	[80]
		A2780 and SKOV3 human ovarian cancer cell lines	[81]
		4T1 and MDA-MB-231 breast cancer cell lines	[82]
	Disruption of mitochondrial membrane potential	A549 and A549/DDP human non-small lung cancer cell lines, SKOV3 and SKOV3/DDP ovarian cancer cell lines	[83]
		HeLa cervical cancer line	[84]
		HCT116 human colon cancer cell line	[85]
		NCI-N87 and AGS gastric cancer cell lines	[86]
	Ca^2+^-overload	HepG2 hepatocellular carcinoma cell line	[87]
		K-562 chronic myelogenous leukemia cell	[88]
Chlorpromazine	Modulation of mTOR-signaling	U-87MG human glioma cells	[89]
		HSC-3 and Ca9-22 human oral cancer cells	[90]
Promethazine	Modulation of AKT-signaling	HT29 and SW480 human colorectal carcinoma cell lines	[91]
		K-562 chronic myelogenous leukemia cell	[92]
Fluphenazine	Modulation of mTOR-signaling	MDA-MB-231 human breast cancer cell line	[93]
	Modulation of AKT-signaling	OVCAR-3 ovarian cancer cell line	[94]

## Data Availability

Not applicable.

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
