# Peer review of "Repurposing Antidepressants and Phenothiazine Antipsychotics as Efflux Pump Inhibitors in Cancer and Infectious Diseases"

_antibiotics, 2023, doi:10.3390/antibiotics12010137_

Round 1

Reviewer 1 Report

Generally, I felt difficult to find the focus of this review paper and got distracted by contents such as "anticancer" and "antidepressant" . To me, the current manuscript reads more like a compile of reports, rather than an interesting review on a specific topic. Secondly, although the English writing is acceptable (some minor changes required), the manuscript is not well-organized. For example, there are several single sentence paragraphs in Pages 4-5 (i.e. lines 156-160, 212-213, etc). 

Author Response

Comment:

Generally, I felt difficult to find the focus of this review paper and got distracted by contents such as "anticancer" and "antidepressant" . To me, the current manuscript reads more like a compile of reports, rather than an interesting review on a specific topic. Secondly, although the English writing is acceptable (some minor changes required), the manuscript is not well-organized. For example, there are several single sentence paragraphs in Pages 4-5 (i.e. lines 156-160, 212-213, etc). 

Answer:

We thank the reviewer for the valuable comments that improved the quality of the manuscript. The single sentence paragraphs have been corrected and rewritten (lines 179-18, 210-211, 228-231, 234-236), the minor errors in English have been corrected.

Reviewer 2 Report

Comments:

The manuscript “Repurposing antidepressants and phenothiazine antipsychotics 2 as efflux pump inhibitors” by Bálint Rácz summarizes the repurposing of SSRIs, TCAs and phenothiazine on anti-multiple drug resistant bacteria and anti-cancers. The subject is interesting for taking advantages of clinical approved drug in several way as the authors mentioned in the manuscript. Overall, it is well written and organized.

However, I am still having some minor concerns on it:

1.       The title seems not matching with the context in the manuscript. As the authors described in the text and tables, these drugs act on MDR bacteria and cancers by multiple mechanisms.

2.       That would be more informative if the authors summarize the dosage of each drug in vitro or vivo in their activity in the table. Because I find that they mentioned that the concentrations of Imipramine and phenothiazines in the assay can not be used in patients in line 182 and 211, respectively.

3.       Line 120-121, how does efflux pump inhibitor alone increase the intracellular concentration of antibiotics, which make me confusing? Probably, the author would like to rewrite the sentence.

4.       I am wondering that potential risks are also applicable described in part of 2.3 to the anti-cancer treatment when using these drugs even we take their advantages of anti-cancer activity.

5.       The authors describe the various mechanisms of three types of drugs in line 257-261 and draw the vivid cartoons in the figure 3. They look inconsistent.

Author Response

Reviewer 2

Comments:

The manuscript “Repurposing antidepressants and phenothiazine antipsychotics 2 as efflux pump inhibitors” by Bálint Rácz summarizes the repurposing of SSRIs, TCAs and phenothiazine on anti-multiple drug resistant bacteria and anti-cancers. The subject is interesting for taking advantages of clinical approved drug in several way as the authors mentioned in the manuscript. Overall, it is well written and organized.

Answer:

We thank the reviewer for the valuable comments that improved the quality of the manuscript.

  1. Comment: The title seems not matching with the context in the manuscript. As the authors described in the text and tables, these drugs act on MDR bacteria and cancers by multiple mechanisms.

Answer: We thank the reviewer for the comment. However, in this review we wanted to focus on the efflux pump inhibiting activities of antidepressants and phenothiazine antipsychotics besides their multiple effects in bacteria and cancer cells.

This article aims to demonstrate the multiple mechanisms of antidepressants and phenothiazine antipsychotics that may contribute to efflux pump inhibition and inhibit MDR bacteria and cancer. MDR in bacteria and cancer is a result of multiple intra- and extracellular mechanisms that are interlinked. The expression and function of efflux pumps are influenced by different mechanisms; thus it may be more reasonable to discuss efflux pumps in a broader context. From our point of view focusing only on the direct efflux pump inhibition in cancer cells by these drugs would be only a pharmacokinetic study. and from a therapeutic approach inhibiting solely the efflux pumps may be ineffective regarding MDR cancers.

  1. Comment: That would be more informative if the authors summarize the dosage of each drug in vitro or vivo in their activity in the table. Because I find that they mentioned that the concentrations of Imipramine and phenothiazines in the assay can not be used in patients in line 182 and 211, respectively.

Answer: The concentration regarding imipramine and phenothiazines is mentioned. In our opinion summarizing in vitro and in vivo plasma concentrations in patients are not informative in the context of this article, since additional pharmacokinetic factors may play a role regarding to the distribution of the drugs. E.g. thioridazine may accumulate intracellularly in macrophages, which is useful in mycobacterial infections. Another problem is that infections and cancers are generally localized to organs and tissues that may be difficult to reach by the given drugs. Additionally, the tissue organization of solid tumors (e.g. neoangiogenesis and poor lymphatic circulation) or abscesses may lead to additional challenges. Therefore discussing in vitro and plasma concentrations are outside of the scope of this review, although it would be important to measure tissue accumulation of these drugs.

Some values have been added to the manuscript referring to the therapeutic dose (reference: Pharmacopsychiatry 2011; 44(06): 195-235, DOI: 10.1055/s-0031-1286287) and the concentration applied in the bacterial assays.

  1. Comment: Line 120-121, how does efflux pump inhibitor alone increase the intracellular concentration of antibiotics, which make me confusing? Probably, the author would like to rewrite the sentence.

Answer: Efflux pump inhibitor alone can sensitize bacterial cells before antibiotic treatment as described in the following paper: doi: 10.4314/ahs.v20i4.16. The sentence has been rewritten.

  1. Comment:  I am wondering that potential risks are also applicable described in part of 2.3 to the anti-cancer treatment when using these drugs even we take their advantages of anti-cancer activity.

Answer: Thank you for your input. Dysbiosis may promote the development of various diseases from obesity to cancer and mental health problems. In our opinion it would be more beneficial to apply a holistic approach also in terms of pharmacological therapy. This means it would be rational to investigate the effects of drugs on various organ systems (e.g. cardiovascular, central nervous system, cancers) and on the microbiome as well. Therefore, it would be important to investigate the effect of cancer therapy and the supportive therapy of cancer patients on the microbiome and correlate with the prognosis of the disease.

  1. Comment: The authors describe the various mechanisms of three types of drugs in line 257-261 and draw the vivid cartoons in the figure 3. They look inconsistent.

Answer: The title of the figure was altered to „Main mechanisms of anticancer activity of SSRIs, TCAs and phenothiazine-type antipsychotics.

Reviewer 3 Report

The topic of this review article, Repurposing antidepressants and phenothiazine antipsychotics as efflux pump inhibitors, is critical. The review article was written, presented, and discussed in a systematic manner. This article accepts its current form.

Author Response

Answer:

We thank the reviewer for the positive feedback. The minor errors in English have been corrected.

Reviewer 4 Report

The manuscript submitted by Rácz et al reports the review on how the antidepressants and antipsychotics drugs can be used as an efflux pump inhibitor for antibacterial and anticancer agents. This manuscript is well structured overall, but it needs little modification that I have outlined in the comments below.

1. Line 41-42: “lack of financial motivation”? – Please use scientific sentence

2. Line 45-46: Please rephrase the sentence

3. Figure 1: Please redraw the figure, it’s hard for the readers to understand (What are purple circles?)

4.  Line 92-93: How this sentence is relevant to the antibacterial resistance?

5.  Line 95: “antibiotic”? - Does the author mean “antibiotic drug”?
6.  Line 219: Mmr? – The authors have to clarify what does it mean

7.  Figure 3: This is not related to the review main idea

8.The authors should add in vitroin vivo IC50/EC50 concentrations of the drugs mentioned in this review.

9. Section 3.1, 3.2 and 3.3: The authors have to explain how these sections are relevant to efflux pump inhibitors?

10. The authors have to draw the structures of some important antidepressants and antipsychotics drugs that are used as an antibacterial and anticancer activity

11.  Section 4: Discussion? I think “Conclusion” sounds better

12.  The authors can add a section, discussing about dual-drug delivery

13.  The authors have to proofread the manuscript, there are grammatical errors

Author Response

Reviewer 4

Comments:

The manuscript submitted by Rácz et al reports the review on how the antidepressants and antipsychotics drugs can be used as an efflux pump inhibitor for antibacterial and anticancer agents. This manuscript is well structured overall, but it needs little modification that I have outlined in the comments below.

Answer: We thank the reviewer for the valuable comments that improved the quality of the manuscript.

  1. Line 41-42: “lack of financial motivation”? – Please use scientific sentence

Answer: The sentence was corrected.

  1. Line 45-46: Please rephrase the sentence

Answer: The sentence was corrected.

  1. Figure 1: Please redraw the figure, it’s hard for the readers to understand (What are purple circles?)

Answer: The figure was corrected and redrawn, the purple circles were removed.

  1. Line 92-93: How this sentence is relevant to the antibacterial resistance?

Answer: The sentence explains the origin of resistance that can be either intrinsic or extrinsic.

  1. Line 95: “antibiotic”? - Does the author mean “antibiotic drug”?

Answer: The word „antibiotic” was changed to „antibiotics”.

  1. Line 219: Mmr? – The authors have to clarify what does it mean

Answer: It is written in the sentence that Mmr is an efflux pump of M. tuberculosis. Mmr efflux pump is present in Mycobacterium smegmatis and M. tuberculosis.

  1. Figure 3: This is not related to the review main idea

Answer: MDR in cancer is a result of multiple intra- and extracellular mechanisms that are interlinked. The expression and function of efflux pumps are influenced by other mechanisms; therefore it may be more reasonable to discuss efflux pumps in a broader context. From our point of view focusing only on the direct efflux pump inhibition by these drugs would be only a pharmacokinetic study. And from a therapeutic approach inhibiting solely the efflux pumps may be ineffective regarding MDR pathogens or cancers.

8.The authors should add in vitroin vivo IC50/EC50 concentrations of the drugs mentioned in this review.

Answer: In our opinion summarizing in vitro and in vivo (plasma?) concentrations (of patients?) are not informative in the context of this article, since additional pharmacokinetic factors may play a role regarding to the distribution of the drugs. E.g. thioridazine may accumulate intracellularly in macrophages, which is useful in mycobacterial infections. Another problem is that infections and cancers are generally localized to organs and tissues that may be difficult to reach by the given drugs. Additionally, the tissue organization of solid tumors (e.g. neoangiogenesis and poor lymphatic circulation) or abscesses may lead to additional challenges. Therefore discussing in vitro and plasma concentrations are outside of the scope of this review, although it would be important to measure tissue accumulation in the studies.

For some drugs the in vitro and in vivo concentrations have been added to demonstrate the difference between laboratory experiments and in vivo concentrations. As indicated in the review the concentrations used for efflux pump inhibition under laboratory conditions are not feasible in the patients.

  1. Section 3.1, 3.2 and 3.3: The authors have to explain how these sections are relevant to efflux pump inhibitors?

Answer: Multidrug resistance in cancer is a result of multiple intra- and extracellular mechanisms that are interconnected. The expression and function of efflux pumps are influenced by other mechanisms; therefore it may be more reasonable to discuss efflux pumps in a broader context.

  1. The authors have to draw the structures of some important antidepressants and antipsychotics drugs that are used as an antibacterial and anticancer activity

Answer: The structures of some important antidepressants and antipsychotics were added as new figures (new Figure 2 and new Figure 3).

  1. Section 4: Discussion? I think “Conclusion” sounds better

Answer: We thank the reviewer for the remark, the title „Discussion” was changed to „Conclusion”.

  1. The authors can add a section, discussing about dual-drug delivery

Answer: We think that the topic of dual drug delivery is out of the scope of this review.

  1. The authors have to proofread the manuscript, there are grammatical errors

Answer: The grammar errors of the manuscript have been corrected.

Reviewer 5 Report

In this article, the authors discuss the role of efflux pumps in infection and the progression of cancer, and the importance of efflux pump inhibitors as a treatment option. In addition, they discuss repurposing the already existing drugs to use as efflux pump inhibitors instead of developing new drugs. They provide an overview of antibacterial and anticancer effects of selective serotonin reuptake inhibitors (SSRIs), tricyclic antidepressants (TCAs) and phenothiazine antipsychotics, and their role as efflux pump inhibitors. Finally, they brief the risks associated with the use of these drugs as efflux pump inhibitors. The manuscript is well-written and nicely organized though I have some minor comments to improve the quality of the manuscript.

Comments:

L141-142: it is confusing as the drug increases the expression of efflux pump genes

L156-159 could be added to the paragraphs above or below as they contain only single sentences

L202: Combine with the paragraph above

L212, L225: Single sentences; please add them to the sentences above or below.

L279: tumorigenic

L314: ER à full form

L334-L336: Please modify or split the sentence

L361: paclitaxel,

L359-L363: Need to split the sentence

L366: model,

L365: P-gp highly expressing à not clear; please re-write

L373-L376: Split the sentence

L376: derivative,

L379: decreasing

L391: virulence, and in cancer,

L398: antipsychotics,

L411: are rising that may be explained à Please re-write it

L411-L414: Please split the sentence

L414-417: Please split the sentence

Author Response

Reviewer 5

Comments:

In this article, the authors discuss the role of efflux pumps in infection and the progression of cancer, and the importance of efflux pump inhibitors as a treatment option. In addition, they discuss repurposing the already existing drugs to use as efflux pump inhibitors instead of developing new drugs. They provide an overview of antibacterial and anticancer effects of selective serotonin reuptake inhibitors (SSRIs), tricyclic antidepressants (TCAs) and phenothiazine antipsychotics, and their role as efflux pump inhibitors. Finally, they brief the risks associated with the use of these drugs as efflux pump inhibitors. The manuscript is well-written and nicely organized though I have some minor comments to improve the quality of the manuscript.

Answer:

We thank the reviewer for the valuable comments that improved the quality of the manuscript.

Comments:

L141-142: it is confusing as the drug increases the expression of efflux pump genes

Answer: Increased expression of marA and acrB was detected after sertraline treatment indicating that this drug induced a stress responses in bacteria resulting in increased expression of efflux pump genes. Similar results were obtained by our former study demonstrating the stress response inducing activity of promethazine in Escherichia coli (Reference: Nové et al. In Vivo. 2020 Jan-Feb;34(1):65-71. doi: 10.21873/invivo.11746.).

L156-159 could be added to the paragraphs above or below as they contain only single sentences

Answer: The sentence was added to the previous paragraph.

L202: Combine with the paragraph above

Answer: The sentence was added to the previous paragraph.

L212, L225: Single sentences; please add them to the sentences above or below.

Answer: The sentences were added to the previous paragraph.

L314: ER à full form

Answer: The term endoplasmic reticulum (ER) is fully written in L268-269, that’ why later only the abbreviated form ER is used.

L334-L336: Please modify or split the sentence

Answer: The sentence was modified.

L361: paclitaxel,

Answer: The sentence was corrected.

L359-L363: Need to split the sentence

Answer: The sentence was modified.

L366: model,

Answer: The sentence was corrected.

L365: P-gp highly expressing à not clear; please re-write

Answer: The sentence was corrected.

L373-L376: Split the sentence

Answer: The sentence was corrected.

L376: derivative,

Answer: The sentence was corrected.

L379: decreasing

Answer: The sentence was corrected.

L391: virulence, and in cancer,

Answer: The sentence was corrected.

L398: antipsychotics,

Answer: The sentence was corrected.

L411: are rising that may be explained à Please re-write it

Answer: The sentence was corrected and rewritten.

L411-L414: Please split the sentence

L414-417: Please split the sentence

Answer: The sentences were corrected.

Round 2

Reviewer 1 Report

The organization has been imporved. The current version reads acceptable.